# Genome-wide association study revealed genomic regions associated with tuber quality traits in water yam (*Dioscorea alata* L.)

Fatoumata Ouattara[1,2]*, Paterne A. Agre[2], Idris I. Adejumobi[2], Bunmi Olasanmi[3], Adekemi Stanley[2], Fatogoma Sorho[4], Konan E. B. Dibi[5], Malachy O. Akoroda[3], Ranjana Bhattacharjee[2]*

**1** Pan African University Life and Earth Science Institute, University of Ibadan, Ibadan, Nigeria, **2** International Institute of Tropical Agriculture (IITA), Ibadan PMB, Nigeria, **3** Department of Crop and Horticultural Sciences, University of Ibadan, Ibadan, Nigeria, **4** African Center of Excellence on Climate Change, Biodiversity and Sustainable Agriculture (CEA-CCBAD), Félix Houphouet-Boigny University, Abidjan Côte d'Ivoire, **5** Centre National de Recherche Agronomique (CNRA), Bouaké, Côte D'Ivoire

* timfa.ouattara@gmail.com (FO), r.bhattacharjee@cgiar.org (RB)

## Abstract

Water yam (*Dioscorea alata* L.) or greater yam, is an essential species of the Dioscoreceae family in tropical and subtropical regions. The wide geographical distribution is owing to its higher tuber yield, storability, and better nutritional and health benefits compared to many other species. Despite these promising characteristics, water yam remains less preferred by consumers for traditional food products, particularly boiled and pounded yam preparations. Fast and efficient development of superior genotypes that meet farmers and end-users needs have been challenging through classical breeding methods. The objective of the study was to use genome-wide associations to assess the genetics of post-harvest tuber quality, mainly targeting the consumer-preferred traits. A panel of 404 water yam genotypes were assessed to decipher the genomic regions associated with traits such as tuber oxidative browning, dry matter content, and boiled and pounded tuber quality. The Multiple Random Mixed Linear Model was employed for marker-trait association analysis using the naive, Q, and Q + K models, followed by gene annotation and marker or allele substitution effects. Fourteen SNP markers were significantly linked with the assessed tuber quality traits and $r^2$ values ranged from 0.62 to 10.02%. The gene annotation analysis revealed presence of 32 putative candidate genes playing crucial roles in enzymatic browning and carbohydrate biosynthesis pathways for dry matter accumulation. The molecular information generated in the present study can be deployed for water yam improvement.

**Data availability statement:** All data supporting the research is available within the manuscript and supplementary files.

**Funding:** The first author is a recipient of PhD research funding from PAN African University of Life & Earth Science Institute (PAULESI) from 2021 to 2024. Financial support was also obtained from Bill & Melinda Gates Foundation by co-author Paterne A. Agre (BMGF, Grant Number OPP1052998). The funders had no role in study design, data collection and analysis, decision to publish, or preparation of the manuscript.

**Competing interests:** No authors have competing interests.

## Introduction

Food quality characteristics are one of the most critical features that determine the degree of acceptability of products by consumers/users, as they directly affect their preferences and perceptions [1]. These characteristics include nutritional value (nutrients and anti-nutritional factors) and organoleptic attributes (texture, appearance, flavor, and aroma) [2].

Yam (*Dioscorea* spp.) is a versatile tuber crop with a unique flavor profile and nutritional benefits [3]. It is a significant source of carbohydrates, dietary fiber, and other essential nutrients for millions of people in the tropical and subtropical regions [4]. It is a perennial crop with herbaceous vine. Its cultivation is mainly for consumption of the starchy tubers in these regions [5]. Yams play a significant role in food security, medicine, and economies of developing nations. Globally, they rank as the fourth most important root and tuber crop after potatoes, cassava, and sweet potatoes. In West Africa, yams are second only to cassava in importance [6]. Yam tubers are consumed in different forms, although the main food products in West African countries are boiled and pounded forms [7]. However, other quality traits such as tuber flesh browning and dry matter content, frequently influence the suitability and acceptance of tubers by end-users [8]. The browning of tuber flesh color in yams after cutting and exposure to air is related with oxidation of polyphenols present in the tubers, and this is associated with modified taste (bitterness) and texture (hardness) [9]. Similarly, high dry matter content in yams is a preferred quality characteristic for producers, processors, and consumers. These preferences have led the end users to favor some yam varieties/species against others regardless of their better agronomic characteristics [10] such as resistance to disease, better yield, etc [11].

Water yam (*Dioscorea alata* L.), also known as greater yam, is one of the most economically significant and geographically widespread yam species and a staple crop in tropical and sub-tropical regions [12]. It has favorable agronomic characteristics and quality attributes, including ease of propagation, early vigor, high yield, better tuber storability [12], low glycemic index [13], antioxidants, as well as high protein, vitamin C, and low lipid contents compared to other economically important *Dioscorea* spp. such as *D. rotundata*, *D. cayenensis*, *D. esculenta* and *D. trifida* [14]. This is in addition to several health benefits, including the anti-inflammatory, anti-cancer, anti-leprosy, and other properties of water yam [15]. Unfortunately, this yam species remains underutilized and unpreferred despite several positive characteristics due to traditional bias and perception among the users about its food quality products compared to widely preferred white yam (*D. rotundata*). Few studies reported development of water yam varieties with superior food quality traits than white yam [16,17]. Other studies targeted specific quality traits such as dry matter content and tuber oxidation browning [18–20]; however, limited information is available on genetic basis of sensory traits such as boiled and pounded characteristics of water yam [21,22]. Genetic improvement of sensory traits, as well as food quality attributes, is an essential target of yam breeding and has become more important over time as end-users are continually demanding cultivars of high quality for traditional and emerging food products [23].

The integration of genomic tools, including genome-wide association study (GWAS), with traditional breeding approaches is a powerful tool for increased efficiency, as genomic regions can be linked to specific traits. GWAS provides higher mapping resolution compared to bi-parental populations in detecting associations between markers and traits of interest. It focuses on assessment of population structure within a panel of genotypes (known as diversity panel) and determines genetic relatedness among individuals by minimizing detection of false associations [24,25]. Recent advances in next-generation sequencing have made it possible to develop and detect many high-quality, genome-wide markers. These markers, including single nucleotide polymorphisms (SNPs), are now used in genomics-based approaches like GWAS to genetically analyse complex quantitative traits in a wide range of crops [26–29].

In recent years, GWAS has been successfully utilized for detecting associations among key agronomic and quality traits in several yam species. These traits include tuber yield and disease resistance [30]; and food product qualities (boiled and pounded tuber characteristics) in water yam [1], and white yam [23]; cooking time, boiled yam hardness, and moldability [22]; tuber dry matter content, tuber flesh color, and oxidative browning [18,31] in water yam; tuber hardness in Bush yam [32]; and yield, plant vigor, and disease resistance in multiple yam species [33]. Although some potential tuber quality and agronomic trait quantitative trait loci (QTLs) have been identified in different *Dioscorea* spp., the genetic studies, particularly focusing on tuber-boiled and pounded characteristics in water yam, are limited. Therefore, this study aimed to use GWAS to pinpoint genomic regions linked to post-harvest tuber quality traits. The specific traits examined included dry matter content, oxidative browning, and boiled and pounded tuber characteristics.

## Materials and methods

### Plant materials

A panel of 404 *D. alata* genotypes with diverse agronomic and tuber quality attributes were selected for the present study (Table 1). These genotypes were selected from the pool of progenies of four different bi-parental populations (using four female and one male parents) developed by Yam Improvement Program (YIP) of the International Institute of Tropical Agriculture (IITA), Ibadan, Nigeria. The four bi-parental populations were combined to carry out GWAS analyses, since the population size of each mapping population was not big enough (Table 1), to make-up the diversity panel. The parents were selected based on their breeding values for traits including tuber yield, dry matter content, and resistance to anthracnose disease. The GWAS panel therefore consisted of 196 progenies and two parents making up the first family/population, 103 progenies and two parents for the second family/population, 54 progenies and two parents for the third family/population, and 43 progenies and two parents for the fourth family/population (Table 1). For field establishment, three mini-setts (approximately 50 g) of each genotype were planted on a 3m one-row plot in an alpha lattice design representing three replications at IITA, Ibadan (7°40'19.62" N, 3°91'73.13" E, and 189 m above sea level), Nigeria. The phenotypic

**Table 1. Description of the planting materials evaluated in Ibadan in two seasons.**

| Mapping Population (Family) | Male parent | Female parent | Number of progenies |
|---|---|---|---|
| Family 1: TDa1419 | TDa02/00012 | TDa99/00240 | 196 |
| Family 2: TDa1427 | TDa02/00012 | TDa95/00328 | 103 |
| Family 3: TDa1403 | TDa02/00012 | TDa00/00005 | 54 |
| Family 4: TDa1402 | TDa02/00012 | TDa05/00015 | 43 |
| Total | | | 396 + 5 parents |

Note: The male parent was common across the four populations. The first two digits in the parental IDs denote the year of crossing. All crosses were performed at IITA. Although the total number of parents used for generating the crosses were five (four female and one male), however, for analyses we considered the sequence information of male and female parent independently for each population (SNP data was different across each male parent for each population, which could be attributed to the clonal maintenance of the crop), resulting in eight parents (four female and four male) and a total panel of 404 genotypes.

evaluation was carried out for two cropping seasons (2021 and 2022). After harvesting, three matured and clean tubers were selected from each genotype, representing three replications, for assessment of quality traits such as dry matter content (DMC), oxidative browning, and tuber sensory evaluation (boiled and pounded characteristics).

## Tuber quality phenotyping

The quality traits were phenotyped using the procedure described by Ouattara *et al.* [34]. In this procedure, for dry matter content, a quantity of 100 g of fresh tuber flesh was grated and dried at 105 °C for 16 h. The percent dry matter content was thereafter determined by using the constant weight as follows:

$$\% \ dry \ matter \ content \ (DMC) \ = \ \frac{\text{Dry tuber flesh weight (g)}}{\text{Wet tuber flesh weight (g)}} \times 100$$

(1)

For tuber oxidative browning, the Chroma meter (CR-400, Konica Minolta, Japan) was used to measure the colour intensity. This was done by recording the L* (lightness), a* (red/green), and b* (yellow/blue) values at 0 and 180 minutes after the tuber was cut. Before each reading, the instrument was calibrated using white and black porcelain tiles. The overall colour difference, or total delta colour difference (ΔE*), was calculated using a specific formula that combined all three-color coordinates as indicated in equations 2 and 3:

$$\Delta E^* = (L^* + a^* + b^*)^{1/2}$$

(2)

$$\text{Oxidative browning (TBOxi)} = \ F\Delta E^* \ - \ I\Delta E^*$$

(3)

where FΔE* is the final delta and IΔE* is the initial delta.

For boiled and pounded traits, healthy tubers from each genotype were peeled and cut into uniform slices. These were then cooked in an electric cooking and pounding machine (QYP-6000, Qasa, Cheerfengly Industrial Co., Ltd., Taipei City, Taiwan) with 380 mL of water for 15–30 minutes. The cooking time was adjusted based on the genotype and tuber texture. The cooked yam was divided into two portions. The first portion was used to assess boiled properties, while the second was immediately pounded using the same machine. Ten IITA-trained staff members, experienced in sensory evaluation, were selected to serve as a panel to assess the genotypes. The specific sensory evaluation attributes for both boiled and pounded quality are detailed in Table 2 and were assessed according to the yam ontology [23].

## Genotyping

DNA from each genotype was isolated at IITA using modified CTAB methods [35] and were genotyped at the Integrated Genotyping Service and Support (IGSS, BecA-ILRI hub, Nairobi, Kenya) using the 'high-density' DArTseq reduced-representation method. The details of sequence length, mapping approach (software used) and other parameters are provided in Bredesson et al [19].

DArTseq genotyping datasets were thereafter mapped onto the v2 genome sequence [19] of water yam (479.5 Mb, 20 chromosomes, ~25,189 protein-coding genes). The raw genotyping data consisted of 19,012 SNPs for family 1 (TDa1419), 17,126 SNPs for family 2 (TDa1427), 13,857 SNPs for family 3 (TDa1403), and 26,643 SNPs for family 4 (TDa1402). The common set of SNPs across the four populations were then searched and a total of 4,805 SNPs were identified for further analyses. The 4,805 SNPs were further subjected to quality control using the parameters of missing values > 20% and minor allele frequency (MAFs) < 0.05, heterozygosity and unmapped SNPs, which resulted in 4,646 high-quality SNPs distributed across the 20 chromosomes of water yam.

**Table 2. Sensory quality attributes (SQA) for boiled and pounded yam tuber quality and scales of rating used by panellists in this study.**

| Traits | Rating scale |
|---|---|
| Boiled tuber quality | |
| Appearance | 1 = Dislike extremely; 2 = Dislike; 3 = Neither like nor dislike; 4 = Like; 5 = Like extremely |
| Color | 1 = Dislike extremely; 2 = Dislike; 3 = Neither like nor dislike; 4 = Like; 5 = Like extremely |
| Aroma | 1 = Dislike extremely; 2 = Dislike; 3 = Neither like nor dislike; 4 = Like; 5 = Like extremely |
| Taste | 1 = Dislike extremely; 2 = Dislike; 3 = Neither like nor dislike; 4 = Like; 5 = Like extremely |
| Texture | 1 = Strong; 2 = Intermediate; 3 = Soft |
| Mealiness | 1 = Soggy; 2 = Slightly mealy; 3 = Mealy |
| Pounded tuber quality | |
| Appearance | 1 = Dislike extremely; 2 = Dislike; 3 = Neither like nor dislike; 4 = Like; 5 = Like extremely |
| Aroma | 1 = Dislike extremely; 2 = Dislike; 3 = Neither like nor dislike; 4 = Like; 5 = Like extremely |
| Color | 1 = Dislike extremely; 2 = Dislike; 3 = Neither like nor dislike; 4 = Like; 5 = Like extremely |
| Mealiness | 1 = Soggy (seedy); 2 = Slightly mealy; 3 = Mealy |
| Mouldability | 1 = not mold well/sticky at hand; 2 = intermediate; 3 = easy to mold |
| Stretchability | 1 = Not elastic/stretch at all; 2 = Intermediate; 3 = Stretch very well or very elastic |
| Taste | 1 = Dislike extremely; 2 = Dislike; 3 = Neither like nor dislike; 4 = Like; 5 = Like extremely |
| Texture | 1 = strong; 2 = intermediate; 3 = soft |

Source: Asfaw *et al.* [23]

## Data analysis

### Phenotypic data

For phenotypic data, a mixed-effects model was used to conduct an analysis of variance (ANOVA), including mean square, coefficient of variance and significance, which was implemented using the lme4 package in the R environment. The traits were considered as dependent while year and genotypes were independent variables. A combined index for the boiled and pounded characteristics was created based on the method described by Ouattara *et al.* [34]. Corrplot R package v0.92 [36] was used to visualize the correlation coefficient generated among the traits.

### Population structure and association mapping (GWAS)

To assess the genetic stratification within water yam panel and the relationship among genotypes, population structure analysis was conducted using three complementary approaches including ADMIXTURE, principal component analysis (PCA), and phylogenetic clustering. To determine the optimal number of genetic cluster (K), ADMIXTURE was run for K value ranging from 1 to 10, and the most appropriate K was identified using the cross-validation (CV) error method implemented in ADMIXTURE model from the LEA package in R [37]. The K value with the lowest CV error was considered the best-supported model. Genotypes were assigned to clusters when their ADMIXTURE membership probability (MP) was ≥ 0.60, and those below this threshold were considered admixed. PCA was performed using factoMiner [38] and factoExtra [39] packages to visualize the distribution of genotypes along major axes of genetic variation, and the clustering pattern from PCA was compared with ADMIXTURE results for consistency. In addition, hierarchical clustering based on pairwise genetic distances (APE package) implemented in R [40,41] was used to further validate the number and membership of clusters.

From the ADMIXTURE model, the Q matrix was obtained. Further, rrBLUP package [42] in R was used to generate the relationship matrix as an important covariate in the GWAS model. Finally, the GWAS analysis was conducted using a

multi-locus random-SNP-effect mixed linear model (mrMLM) in mrMLM package [43] implemented in R. The GWAS model was based on equation 4:

$$y = Xb + Zu + e \tag{4}$$

where y is the vector of the phenotypic value of each trait, X is the matrix of effect (population structure), b is the coefficient of the fixed effects, Z is the design matrix linking random effect to phenotypes, u is the random genetic effects, and e is the residual error.

For GWAS analysis, three different approaches were used. A naive model characterized by zero control for false positive error, Q model characterized by the inclusion of Q-matrix from population structure analysis to minimize false positives, and a Q+K model characterized by the inclusion of Q-matrix and K-matrix from kinship to control false positives was considered. The traits association analysis was conducted using the Best Linear Unbiased Estimates (BLUEs) for the two years of evaluation. Significant SNP markers were detected using the LOD of three, as identified by the adjusted false discovery rate (FDR) for the six genetic models considered in the study. These models were multi-locus random-SNP-effect Mixed Linear Model (mrMLM); Fast multi-locus random-SNP-effect efficient mixed model association (FASTmrEMMA); Iterative Sure Independence Screening EM-Bayesian least absolute shrinkage and selection operator (ISIS EMBLASSO); polygenic-background-control-based least angle regression plus empirical Bayes (pLARmEB); polygenic-background-control-based KruskalWallis test plus empirical Bayes (pKWmEB); and fast mrMLM (FASTmrMLM) [40]. The model fit was further compared using Akaike Information Criterion/Bayesian Information Criterion (AIC/BIC) to identify which model consistently provided the best fit across traits. To ensure robust detection of trait-marker associations, a SNP was considered significant only when it exceeded the threshold LOD (FDR-adjusted), and equally identified by at least two of the six multi-locus methods under the same correction model.

The heatmap displaying the pairwise LD measurements between significant SNPs associated with traits of interest was generated using the Gapit [44] function and visualized with LD heatmap package in R [45].

### Candidate genes annotation

All candidate putative genes within 1MB window (500 kb downstream to 500 kb upstream) of the detected loci were identified. The Yam database (https://yambase.org/organism/Dioscorea_alata/genome; Dioscorea alata Genome Browser at JGI (accessed on March 2024)) was used to identify the expression data and gene annotations. The physical locations of the genes and SNPs were based on *D. alata* V2_Reference genome (*Dioscorea alata* v2.1 DOE-JGI, http://phytozome.jgi.doe.gov/). The candidate genes' annotation functions and related information were obtained from the INTERPRO website (https://www.ebi.ac.uk/interpro/) (last accessed on March 2024).

## Results

### Variation and correlation among phenotypic traits

Significant differences (p < 0.001) were observed among the genotypes for the studied traits, including tuber flesh browning, dry matter content, and boiled and pounded quality (Table 3). Tuber flesh browning (TOxB) had a significant and positive relationship with boiled tuber quality (BldT) (r = 0.28, *p* < 0.001). Similar relationship existed between boiled (BldT) and pounded tuber (PndT) quality (r = 0.48, *p* < 0.001) (Fig 1). In addition, a negative and significant relationship existed between dry matter content (DMC) and oxidative browning (ToxB) (r = −0.27, *p* < 0.001), as well as between DMC and boiled tuber (BldT) quality (r = − 0.21, *p* < 0.001).

### Population structure

Based on CV error curve, K = 3 represented minimum prediction error, and therefore three genetic clusters were retained (Table 4; Fig 2). These genetic clusters comprised 296 water yam genotypes representing 73.27% of the total number of

**Table 3. Analysis of variance (ANOVA) for the sensory traits (pounded yam and boiled yam) in *D. alata*.**

| (A) | Df | TOxB | DMC | BldT | PndT |
|---|---|---|---|---|---|
| Year | 1 | 1645.18*** | 2.66$^{ns}$ | 4.73* | 11.76** |
| Genotypes | 403 | 154.51*** | 71*** | 2.7*** | 2.18*** |
| Year*genotypes | 403 | 62.44*** | 17.96*** | 2.08*** | 1.76*** |
| Residuals | 2032 | 13.02 | 3.23 | 0.23 | 0.36 |
| Mean | | 15.9 | 27.48 | 3.34 | 5.01 |
| Min | | 2.15 | 12 | 1.26 | 2.63 |
| Max | | 29.75 | 37.08 | 5.5 | 6.89 |
| CV (%) | | 30.59 | 9.04 | 27.94 | 27.62 |

TOxB: tuber oxidative browning; DMC: dry matter content; BldT: boiled tuber index; PndT: pounded tuber index

genotypes under study while remaining 108 genotypes (26.73%) were admixed. The details of cluster composition and admixed genotypes is presented in Table 4 and Fig 2. Cluster 1 didn't represent any genotypes or admixed population from Family 1 and Family 2, although 5.7% of genotypes from Family 2 were admixed. The remaining two clusters represented genotypes and admixed populations from all four families (Table 4). As ADMIXTURE, the hierarchical cluster analysis, represented three genetic groups (Fig 3). Cluster 1 (red) had the largest number of genotypes (52.47%), followed by Cluster 2 (blue) (24.75%) and Cluster 3 (green) (22.78%). The first cluster consisted of individuals from families F1 (191), F2 (13), F3 (4), and F4 (4). The second cluster consisted of individuals from families F1 (7), F2 (51), F3 (27), and F4 (15). Similarly, the third cluster consisted of individuals from families F2 (41), F3 (25), and F4 (26). Complementary analysis using the principal component analysis (Fig 4) also grouped the genotypes into three clusters, with the first two components explaining 33.5% of the variation.

### Genome-wide association scan for tuber quality traits

The genome-wide association scan identified several SNPs for all tuber quality traits using the three models (naive, Q, and Q+K) in this study. A SNP was considered significantly associated with a trait only when it was detected by at least two of the six multi-locus methods. For TOxB, two SNPs were detected on chromosomes 8 (Chr8_508116) and 18 (Chr18_3861135), accounting for 2% and 2.87% of the total phenotypic variations, respectively (S1 Table; Fig 5A) using the naïve model. In the Q model, three SNPs were identified on chromosomes 5, 8, and 18, with SNPs on Chr5_1585607, Chr8_508116 (same as naïve model), and Chr18_1822019 accounting for 2.55%, 1.11%, and 2.14% of the phenotypic variations, respectively (S1 Table; Fig 5B). The Q+K model identified four SNPs, comprising three earlier identified SNPs (in the Q model) and one additional SNP on chromosome 18 (Chr18_17075749). The SNPs Chr5_1585607, Chr8_508116, Chr18_1822019, and Chr18_17075749 accounted for 2.02%, 1.11%, 2.14%, and 1.27% of the total phenotypic variations, respectively (S1 Table; Fig 5C). A total of five SNPs were associated with tuber oxidation browning.

For DMC, seven SNPs on chromosomes 1, 5,14, and 20 were linked to the trait. Six SNPs (Chr1_26562068, Chr1_26788459, Chr1_26288741, Chr5_22193453, Chr5_22344109, and Chr14_19089185) were identified using the naïve model contributing to the phenotypic variation ranging from 0.00% to 13.09% (S2 Table; Fig 6A). The Q model identified five SNPs (four SNPs as in naïve model and one on Chr20_16607018) (S2 Table; Fig 6B), while the Q+K model identified six SNPs (as identified in naïve and Q model) with significant associations. The phenotypic variation varied from 0.62% (Chr5_22344109) to 10.02% (Chr14_19089185) (S2 Table; Fig 6C).

A total of six SNP markers identified on chromosomes 3, 15, 17, and 19 were linked with the boiled tuber quality (S3 Table) of which only one SNP marker (Chr19_3840797) was common across all three models. The SNP marker, Chr19_3840797, explained 2.58% (naïve model), 3.36% (Q model), and 3.33% (Q+K model) of the total phenotypic

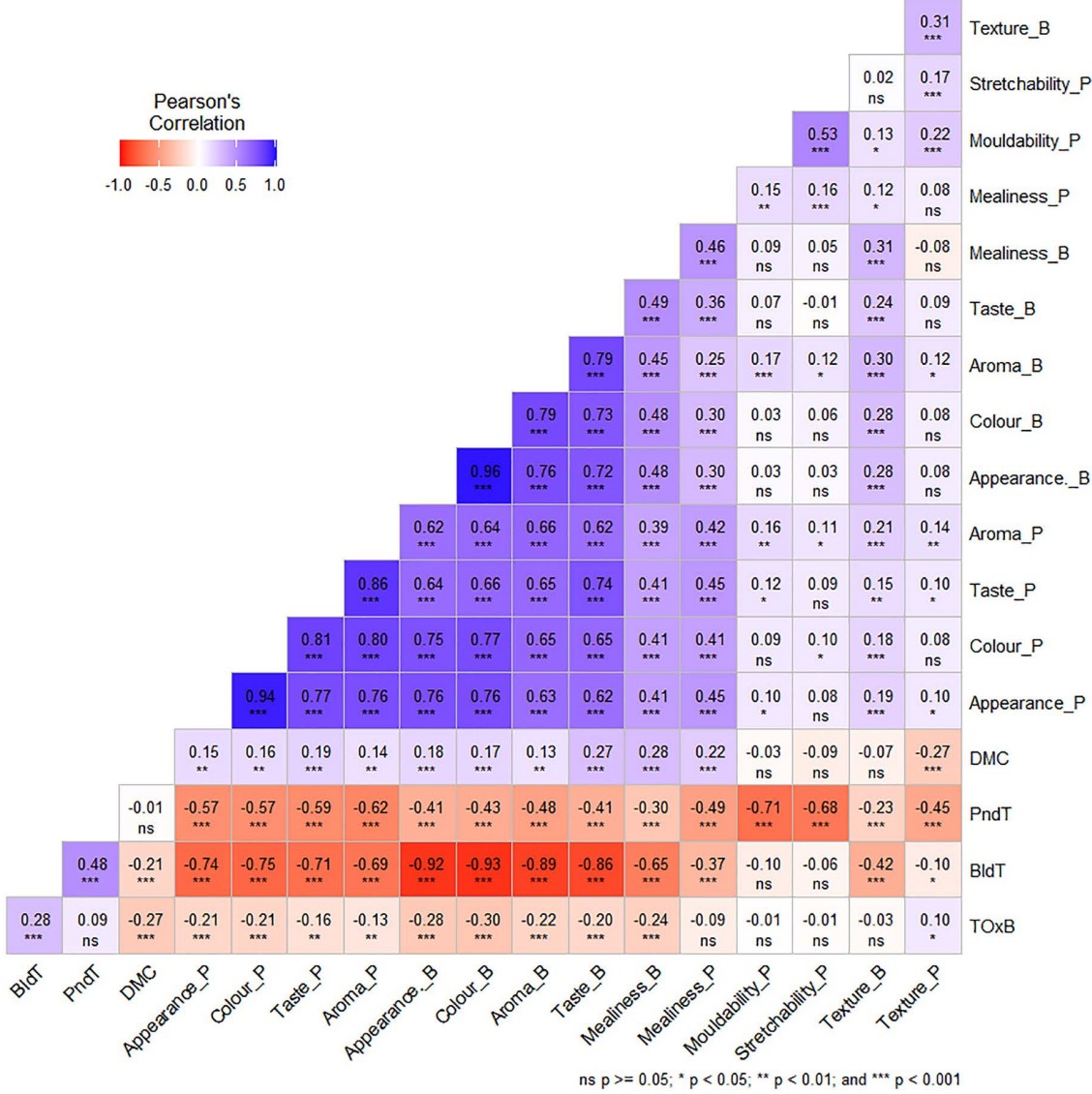

**Fig 1. Correlation coefficients between quality traits and sensory attributes associated with boiled and pounded tuber characteristics.** TOxB: tuber oxidative browning; DMC: dry matter content; BldT: boiled tuber quality index; PndT: pounded tuber quality index; _B: sensory attributes of boiled tuber; _P: sensory attributes of pounded tuber.

variation. The SNP Chr17_522272 was detected by naive and Q+K models and accounted for 2.54% of phenotypic variation. The Q and Q+K models detected SNP, Chr17_434713, accounting for 2.08% of phenotypic variation. The remaining three SNP markers, Chr3_753640, Chr15_22094249, and Chr19_23975255, were identified by the naive model only and explained 0.51%, 2.06% and 1.33% of phenotypic variation, respectively (S3 Table; Fig 7).

For pounded tuber quality, three SNP markers (Chr2_21553389, Chr8_9354121, and Chr14_21319040) were identified, of which SNP Chr14_21319040 was detected in all three models (naive, Q, and Q+K models) and accounted for 3.61% of the total phenotypic variation (Fig 8; S4 Table). The other two SNPs (Chr2_21553389 and Chr8_9354121) were identified by naive model only, which accounted for 1.5% and 3.47% of the phenotypic variation, respectively.

**Table 4. Distribution of 404 genotypes in different clusters from each population/family based on population structure analysis.**

| Population Structure | Population 1 | Population 2 | Population 3 | Population 4 | Number of genotypes per cluster |
|---|---|---|---|---|---|
| Cluster 1 | 0 | 0 | 32.1% (18) | 37.8% (17) | 8.7% (35) |
| Cluster 2 | 41.4% (82) | 33.3% (35) | 16.1% (9) | 2.2% (1) | 31.4% (127) |
| Cluster 3 | 40.9% (81) | 42.9% (45) | 8.9% (5) | 6.7% (3) | 33.2% (134) |
| Cluster 1 Admix | 0 | 5.7% (6) | 26.8% (15) | 37.8% (17) | 9.4% (38) |
| Cluster 2 Admix | 8.6% (17) | 10.5% (11) | 7.2% (4) | 11.1% (5) | 9.2% (37) |
| Cluster 3 Admix | 9.1% (18) | 7.6% (8) | 8.9% (5) | 4.4% (2) | 8.1% (33) |
| Total | 198 (196 progenies + 2 parents) | 105 (103 progenies + 2 parents) | 56 (54 progenies + 2 parents) | 45 (43 progenies + 2 parents) | 404 |

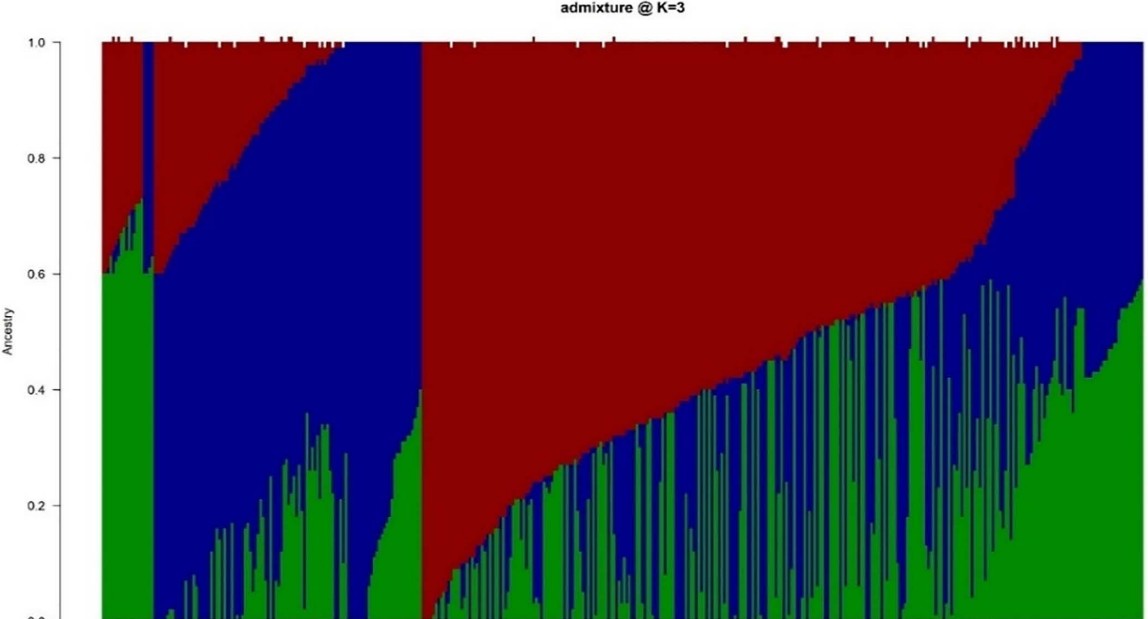

**Fig 2. Population structure of 404 water yam genotypes (k = 3).** Each bar represents one genotype, while each color stands for a group.

## Gene annotation and candidate genes associated with identified SNPs

The SNP markers identified using only false positive control models (Q and Q + K) were considered for gene annotations. Candidate genes prediction was based on linkage disequilibrium decay ($r^2 = 0.1$). Thirty-five putative candidate genes were identified, and their detailed descriptions are summarized in Table 5.

## Candidate genes associated with tuber oxidative browning (TOxB)

A total of eight candidate genes were identified for TOxB located on chromosomes 5, 8, and 18 (Table 5, S5 Fig). These genes include AP2 domain (IPR001471), Cytochrome P450 (IPR001128), Protein kinase domain (IPR000719), Serine-threonine/Protein tyrosine kinase (IPR001245), Glycosyl hydrolase family 1 (IPR001360), Glucose-6-phosphate dehydrogenase C-terminal domain (IPR022675), Phosphomethylpyrimidine Kinase/

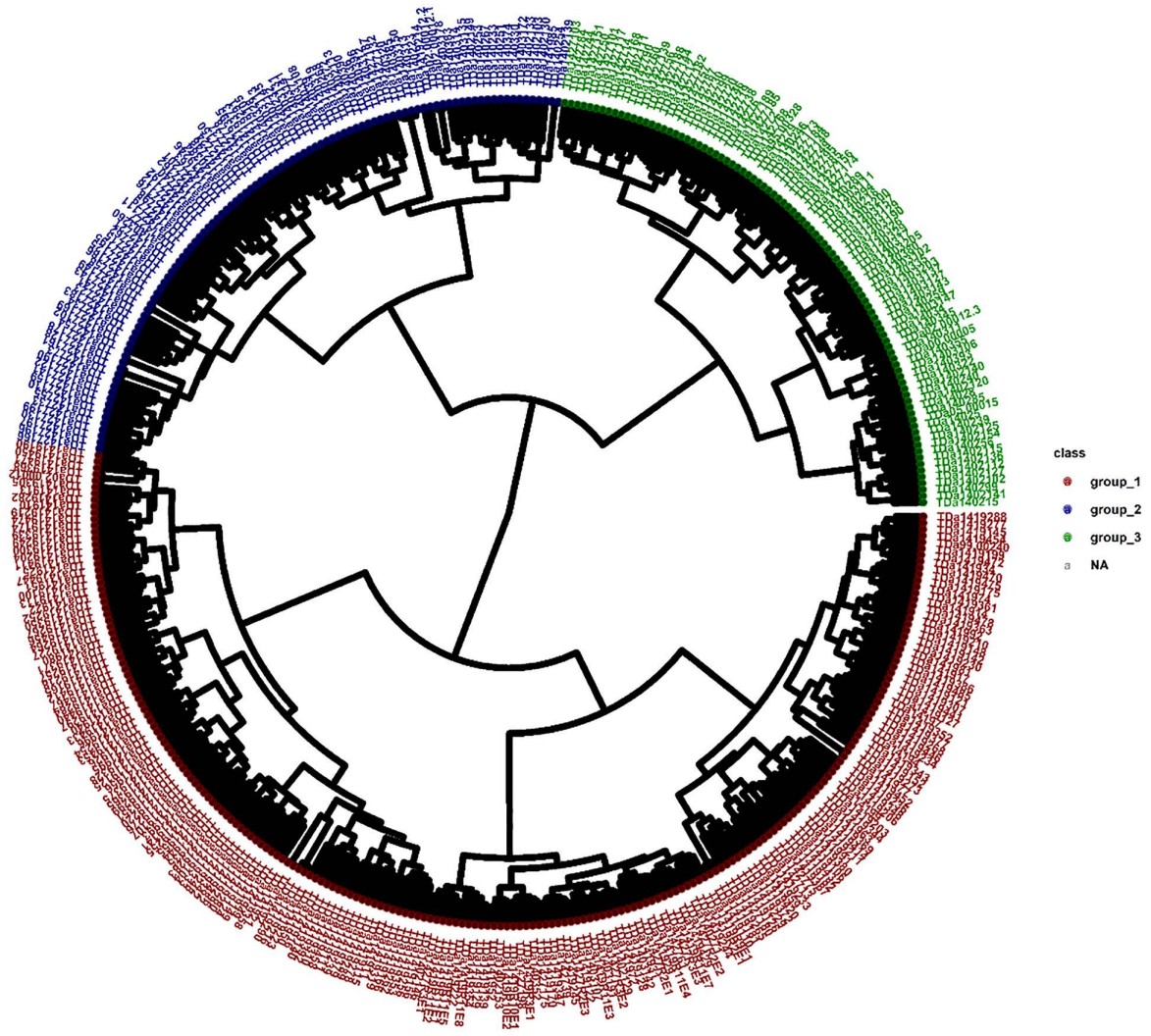

**Fig 3. Hierarchical cluster analysis among 404 water yam genotypes.** Each color represents a genetic group.

Hydroxymethylpyrimidine Phosphokinase (IPR013749), and Thioredoxin (IPR013766). The gene function in oxidative browning has been provided in Table 5.

### Candidate genes linked with dry matter content (DMC)

Sixteen putative candidate genes were identified for DMC on chromosomes 1, 5, 14, and 20 (Table 5, S6 Fig). The SNPs on Chromosome 1 (Chr1_26562068 and Chr1_26788459) were associated with Nucleoside phosphorylase domain and Tetratricopeptide-like helical domain, while the SNPs on Chromosome 5 (Chr5_22193453 and Chr5_22344109) were linked with glycosyltransferase family 29, beta-ketoacyl, lectin C-type domain, AMP-binding enzyme C-terminal domain, Serine/threonine-protein kinase, and glycosyl hydrolase family 9. Similarly, the SNP marker on chromosome 14 is associated with glyceraldehyde 3-phosphate dehydrogenase, tetratricopeptide repeat, protein phosphatase 2A regulatory B subunit, and glycosyltransferase family 17. On the other hand, Thioredoxin domain, UDP-glucuronosyl/UDP-glucosyltransferase, Glycoside hydrolase 35, and catalytic domain were linked with the SNP (Chr20–16607018) on chromosome 20.

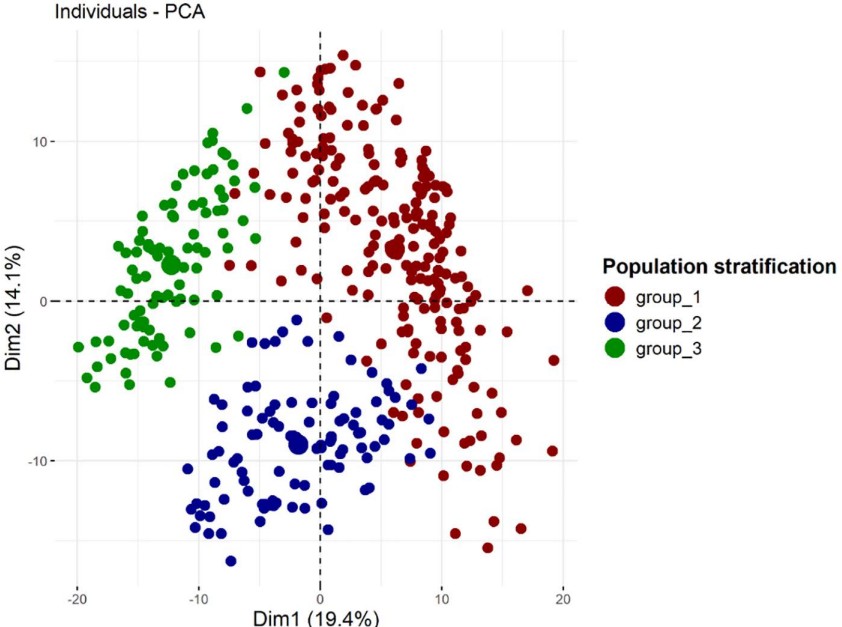

**Fig 4. Dimensional reduction analysis on 404 water yam genotypes using 4,646 SNP-based markers.** Each dot represents an individual genotype.

## Candidate genes associated with boiled tuber (BldT) quality

For BldT quality, two candidate genes were identified on chromosome 17 and 19, respectively. The SNP on chromosome 17 (Chr17_522272) is associated with protein tyrosine kinase, UDP-glucoronosyl and UDP-glucosyl transferase, phospho-glycerate kinase, and cytochrome P450, while the SNP on chromosome 19 (Chr19_3840797) is linked with tetratricopeptide repeat (Table 5, S7 Fig).

## Candidate genes associated with pounded tuber (PndT) quality

For PndT quality, five putative genes were associated with SNPs located on chromosome 14 (Table 5, S8 Fig). These genes are protein kinase domain, protein tyrosine kinase, ubiquitin carboxyl-terminal hydrolase, aldose 1-epimerase, and kelch repeat type 1.

## Discussion

Food quality traits in water yam (*Dioscorea alata* L.) are critical to its wider acceptance by end users, making it a major objective in the crop improvement program. In West Africa, tuber quality traits such as flesh color, boiled and pounded tuber quality play a very important role as far as consumer acceptance is considered, and these traits are associated with the dishes and locations within the region. This complicates the breeding efforts and selection of varieties targeting specific consumer preferences. Several studies have been carried out in white yam, reporting variations in quality traits [10,83]. Very few studies are currently available that addressed tuber quality and sensory traits in water yam. Yams, in general, present a highly complex and heterozygous genome, posing challenges to understand genetic mechanisms behind the traits. This necessitates the need to employ molecular approaches for genetic improvement in yams. GWAS has emerged as one of the powerful tools for identifying key genes associated with targeted agronomic and quality traits. The application of GWAS in yams (both white and water yam) has also become possible due to the availability of high-quality reference genomes.

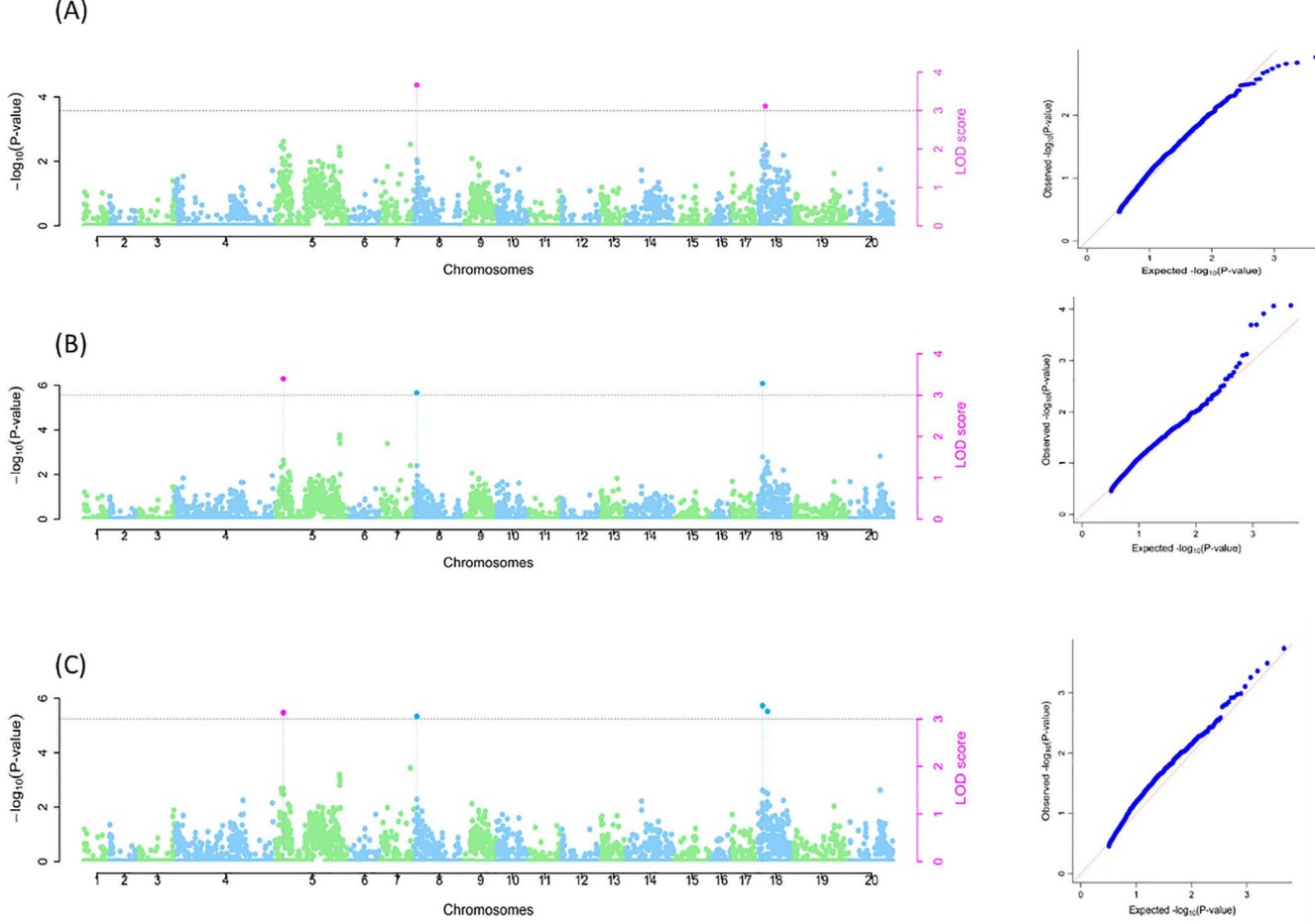

**Fig 5. SNP markers linked with tuber oxidative browning (TOxB).** A: Naïve model; B: Q model; C: Q+K model. The graphs are the Manhattan and Q-Q plots. Grey dotted lines correspond to the threshold (LOD = 3). Pink SNP hit: identified by all six multi-locus methods; Blue SNP hit: identified by at least two of the six multi-locus models.

In this study, we analyzed 404 water yam genotypes originating from four bi-parental populations. All four bi-parental populations were combined for GWAS analyses, as it is well known that population size plays a critical role in potential false positives and negatives in the detection of QTLs underlying targeted traits, which is more obvious in the case of quality traits [84]. The combination of four mapping populations not only took care of population size but also represented a diverse set of genotypes as the panel for better accuracy of GWAS analysis in this study. Furthermore, these populations were derived by crossing four female and one male genotype with good genetic merit for yield, dry matter content, and resistance to anthracnose disease [19]. The panel was expected to be enriched with both major and minor effect alleles underlying the traits under study, and for efficient marker-trait associations [85].

Large phenotypic variation was observed for the four quality traits across 404 water yam genotypes in this study. Furthermore, a strong relationship existed among assessed traits. Interestingly, a strong positive relationship existed between tuber oxidative browning and boiled tuber quality as well as between boiled and pounded tuber quality. A similar positive correlation among boiled and pounded tuber quality was reported in white yam by Asfaw *et al.* [23] and water yam by Ouattara *et al.* [34]. In fact, some of the preferred traits among consumers in yams are white flesh color and better poundability

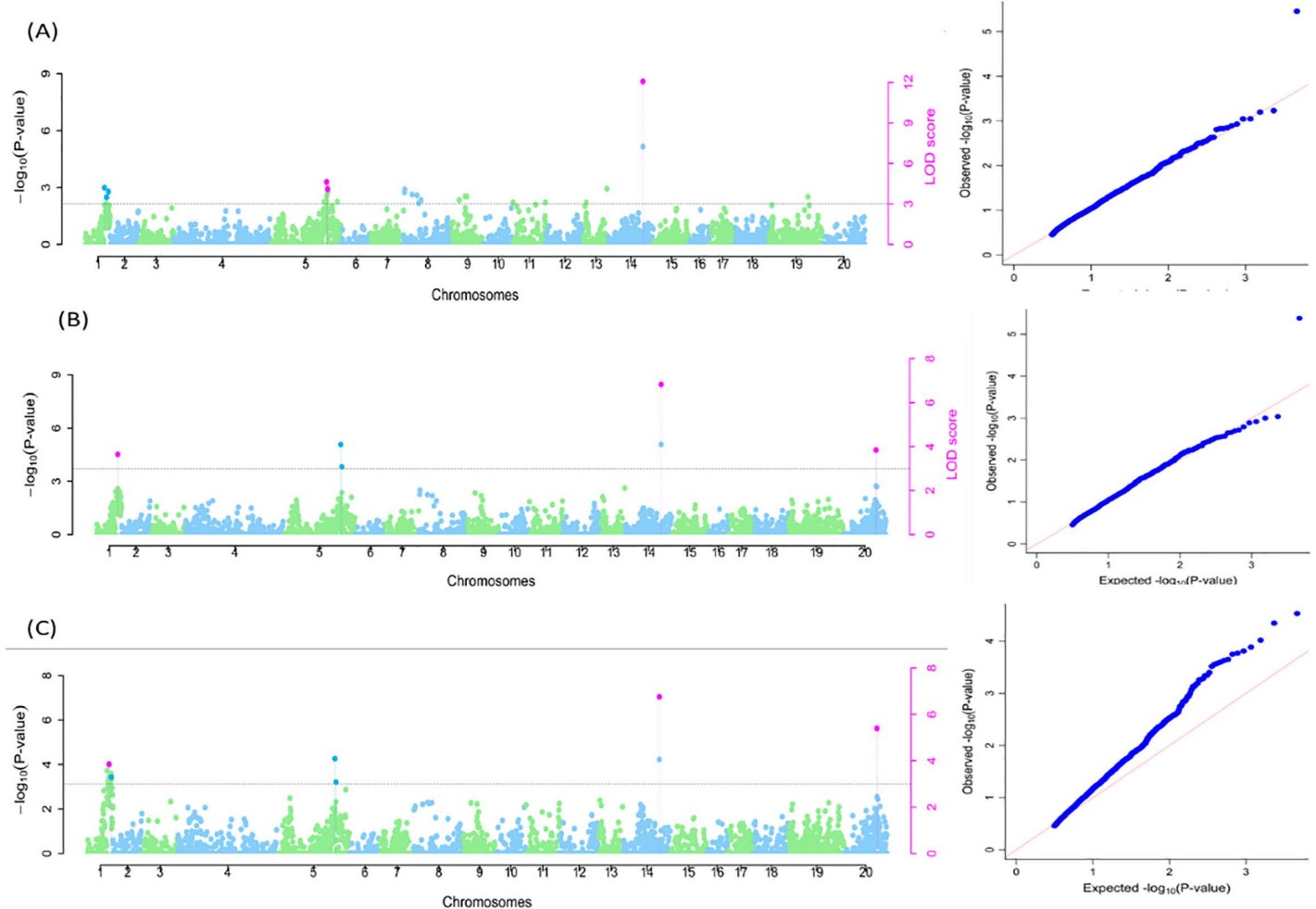

**Fig 6. SNP markers associated with dry matter content.** A: Naïve model; B: Q model and C: Q + K model. The graphs are the Manhattan and Q-Q plots. Grey dotted lines correspond to the threshold (LOD = 3). Pink SNP hit: identified by all six multi-locus methods; Blue SNP hit: identified by at least two of the six multilocus models.

after boiling, based on certain dishes prepared in West Africa, which affects the acceptability of those dishes [7]. In general, oxidative browning affects the visual appeal and taste (bitterness) of yams, making them less attractive to consumers. The poundability of yam depends on tuber dry matter content, and Gatarira et al. [18] reported that yams with high DMC have better eating characteristics such as texture, taste, nutritional value, and good processing features. Similarly, the negative correlation observed between DMC and boiled tuber quality depicts that these two traits are genotype-dependent, which again depends on water absorption rate of genotypes during boiling [86]. The observed negative correlation can be explained due to both linkage disequilibrium and pleiotropy as described by Chen and Lubberstedt [87].

The study identified three genetic subgroups from admixture and PCA analyses with limited genetic diversity, which could be because of the nature of the population panel used in the study. Bredeson et al. [19] highlighted the close relationship between the parents used to generate these populations. They reported that the four female parents shared a grandparent-parent-grandchild relationship and could probably be also related to the male parent. Building on the work of Garin et al. [88], who noted that QTL effects from multi-parent populations can vary based on parentage, we employed six

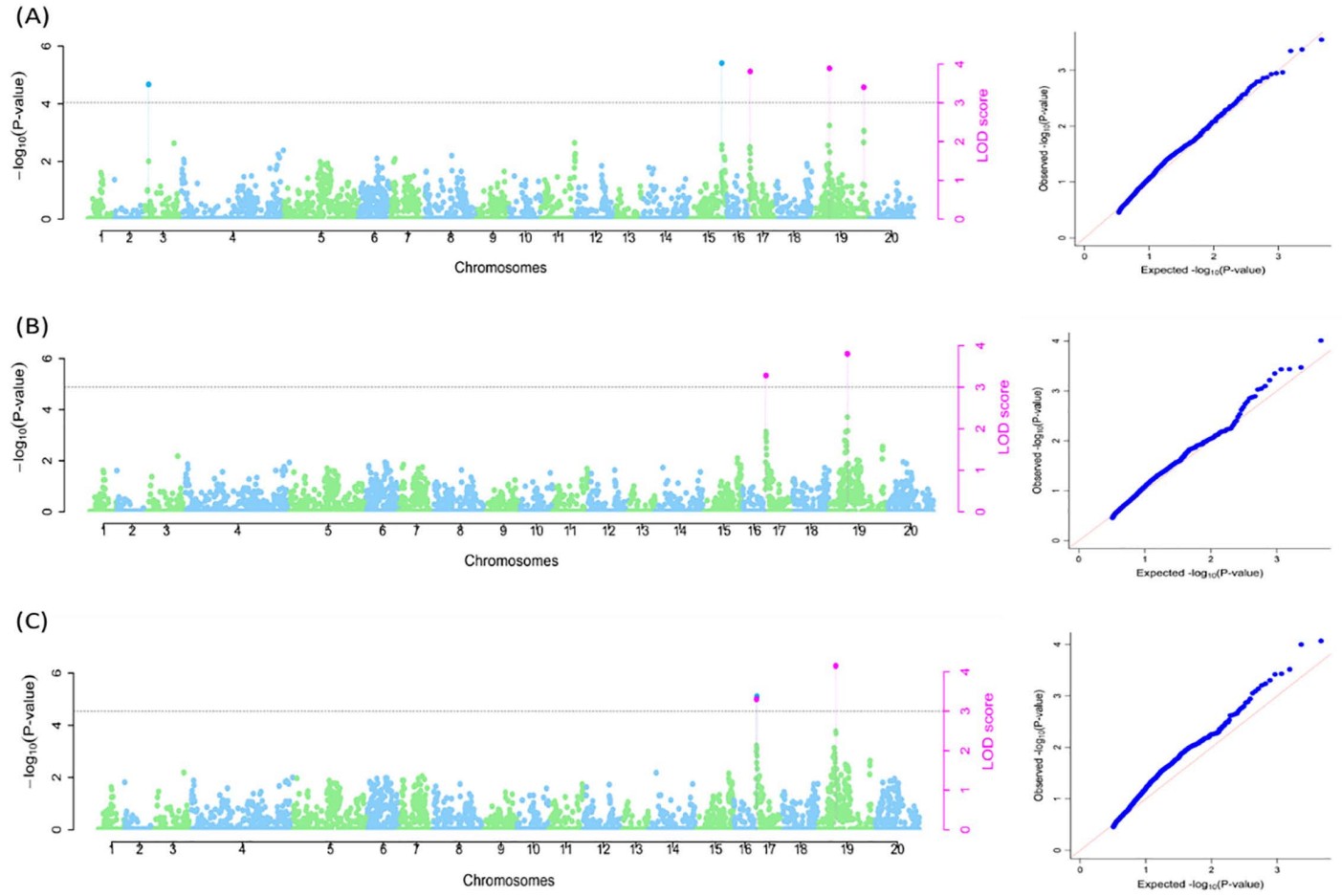

**Fig 7. SNP markers linked with boiled tuber quality.** A: Naïve model; B: Q model and C: Q+K model. The graphs are the Manhattan and Q-Q plots. Grey dotted lines correspond to the threshold (LOD = 3). Pink SNP hit: identified by all six multi-locus methods; Blue SNP hit: identified by at least two of the six multi-locus models.

different genetic models within the mrMLM GWAS model. This allowed us to more accurately identify SNPs associated with key tuber quality traits (dry matter content, oxidative browning) and sensory traits (boiled and pounded quality). Each method detected a different number of SNPs, reflecting their unique power to identify associations.

In this study, GWAS analysis identified more than 20 SNPs associated with four quality-related traits. However, only 14 were detected to be significant using the Q and Q+K models. This is the reason we used a likelihood comparison (AIC/BIC) among the naïve, Q, and Q+K models to demonstrate that the Q+K model provided the best fit for all traits, thereby supporting its primary use in interpreting significant associations. Wang *et al.* [89] reported that SNP markers detected by using false positive control model are more reliable than those identified using the naive model, and, therefore, the SNP identified by the false positive control presented an insight into the genetic control of the targeted traits. For tuber oxidative browning, three SNP markers were identified on chromosomes 5, 8, and 18, which is in line with previous studies [18,19,90], although the physical positions of the SNPs are different. Similarly, for dry matter content, seven SNPs were observed on chromosomes 1, 5, 14, and 20, which are different from those previously reported by Gatarira *et al.* [18], Bredesson *et al.* [19], and Adewumi *et al.* [32]. For tuber-boiled and pounded characteristics, SNPs found on chromosomes 17, 19 (tuber-boiled), and

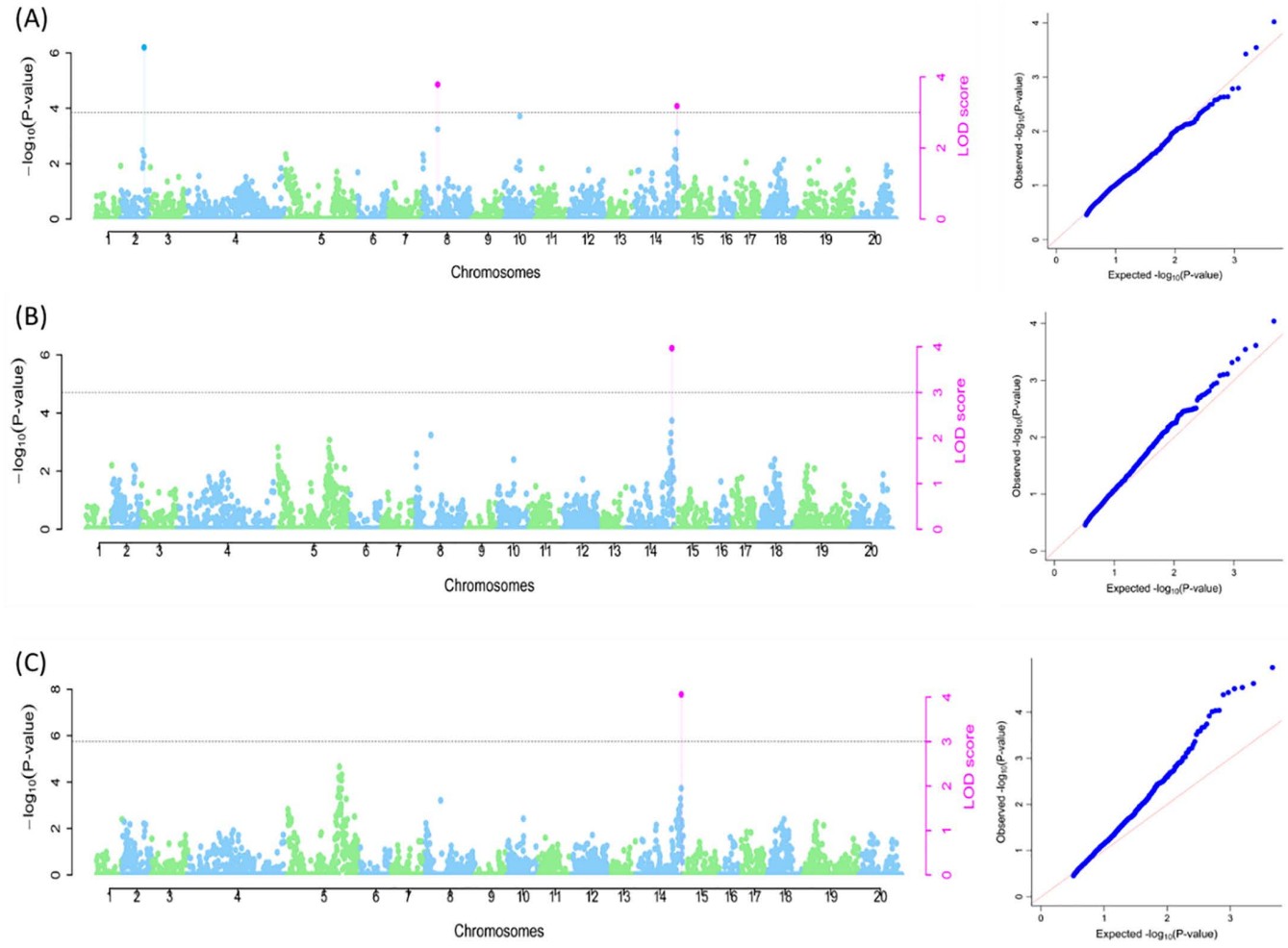

**Fig 8. SNP markers associated with pounded tuber quality.** A: Naïve model; B: Q model and C: Q + K model. The graphs are the Manhattan and Q-Q plots. Grey dotted lines correspond to the threshold (LOD = 3). Pink SNP hit: identified by all six multi-locus models; Blue SNP hit: identified by at least two of the six multi-locus models.

14 (tuber-pounded) were also different from those reported by Dossa *et al.* [22] and Asfaw *et al.* [23] on water yam and white Guinea yam, respectively. The difference between SNPs observed in this study and those found in previous studies in water yam or white Guinea yam could be explained by the complexity in the inheritance of these traits, presence of genotype-by-environment effect, differences in individual genotypes/species, and differences in population size.

Several candidate genes have been pinpointed in this study, including a total of eight putative genes identified for tuber oxidative browning. Among these genes, Cytochrome P450 (IPR001128) enzymes are known for their involvement in oxidations, which could be related to the breakdown of phenolic compounds and other substrates present in water yam tuber that results in oxidative browning [46]. Similarly, Glycosyl hydrolase family 1 is known for hydrolysis of glycosidic bond between two or more carbohydrates [53], resulting in browning reactions [56]. AP2 domain contributes to the regulation of oxidative browning by interacting with gene promoters and influencing histone modifications, controlling gene expression, and conditioning oxidative browning [56,57]. Furthermore, Thioredoxin enzymes play the role of regulating redox balance and protecting against oxidative stress [59,63].

**Table 5. Putative candidate genes associated with tuber quality and sensory traits of water yam (*D. alata*).**

| Markers | Traits | Search position | GO/Interpro ID | Gene family name | Putative gene |
|---------|--------|-----------------|----------------|------------------|---------------|
| Chr5_1585607 | TOxB | 1085607-2085607 | IPR001128, GO:0016705 | Cytochrome P450 | Oxidoreductase activity, acting on paired donors, with incorporation or reduction of molecular oxygen [46,47]. |
| | | | IPR000719, GO:0004672 | Protein kinase domain | A key regulatory component that allows tissues to control enzymatic browning by phosphorylating and activating the enzymes involved and integrating various signaling inputs like calcium that coordinate the browning response [48,49]. |
| | | | IPR001245, GO:0004672 | Protein tyrosine kinase | Catalyses the transfer of the gamma phosphate from nucleotide triphosphates (often ATP) to one or more amino acid residues in a protein substrate side chain, resulting in a conformational change affecting protein function [50,51]. |
| | | | IPR001360, GO:0004553 | Glycoside hydrolase family 1 | Hydrolyse the glycosidic bond between two or more carbohydrates or between a carbohydrate and a non-carbohydrate moiety [52,53]. |
| | | | IPR022675, GO:005514 | Glucose-6-phosphate dehydrogenase, C-terminal domain | Catalyses the first step in the pentose pathway (PPP) and plays an essential role in the oxidative stress response by producing NADPH, the main intracellular reductant [54]. |
| | | | IPR013749, GO:0009228 | Phosphomethylpyrimidine Kinase | Serves as a cofactor for many enzymes that are involved in amino acid and sugar metabolism [55]. |
| | | | IPR001471, GO:0003700 | AP2 domain | Contributes to the regulation of browning by interacting with gene promoters, responding to DNA methylation changes, and influencing histone modifications that control gene expression during the browning process in fruits and tubers [56,57]. |
| Chr8_508116 | TOxB | 8116-1008116 | IPR001128 | Cytochrome P450 | |
| | | | IPR000719, GO:0004672 | Protein kinase domain | |
| Chr18_1822019 | TOxB | 1322019-2322019 | IPR013766, GO:0045454 | Thioredoxin | Regulates the redox balance and protects against oxidative stress [58–60]. |
| Chr1_26562068 | DMC | 26062068-27062068 | IPR000845, GO:0009116 | Nucleoside phosphorylase domain | Catalyses the cleavage of guanosine or inosine to respective bases and sugar-1-phosphate molecules [61,62] |
| Chr1_26788459 | DMC | 26288459-27288459 | IPR011990, GO:0005515 | Tetratricopeptide-like helical domain | The TPR motif interacts directly with isoamylase 1 (ISA1), a key enzyme involved in starch synthesis [63]. |
| Chr5_22193453 | DMC | 21693453-22693453 | IPR001675, GO:0006486 | Glycosyltransferase family 29 | Responsible for synthesizing the glycosidic linkages that form the structure of starch molecules, contributing to the overall starch content in plant tissues [64,65]. |
| | | | IPR013747, GO:0006633 | beta-ketoacyl-[acyl-carrier-protein] synthase III | The enzyme responsible for initiating the chain of reactions of the fatty acid synthase in plants [66]. |
| | | | IPR001304, GO:0004672 | Lectin C-type domain | Involved in recognizing and binding to carbohydrates in various biological contexts, including plant metabolism [67]. |
| | | | IPR025110, GO:0003824 | AMP-binding enzyme C-terminal domain | Facilitating the binding of AMP (adenosine monophosphate) to specific enzymes involved in carbohydrate metabolism [68,69]. |
| | | | IPR001245, GO:0004672 | Serine/threonine-protein kinase | Plays a key role in maintaining energy homeostasis by post translationally regulating various enzymes involved in carbohydrate metabolism pathways [70,71]. |
| | | | IPR001701, GO:0004553 | Glycosyl hydrolase family 9 | Involved in diverse enzymatic metabolisms of carbohydrate compounds available in many plant tissues [53]. |
| Chr5_22344109 | DMC | 21844109-22844109 | IPR001701, GO:0004553 | Glycosyl hydrolase family 9 | |

*(Continued)*

**Table 5.** (Continued)

| Markers | Traits | Search position | GO/Interpro ID | Gene family name | Putative gene |
|---|---|---|---|---|---|
| Chr14_19089185 | DMC | 18589185-19589185 | IPR020829, GO:0016620 | Glyceraldehyde 3-phosphate dehydrogenase, C-terminal domain | Plays a crucial role in carbohydrate metabolism, starch accumulation, and sucrose synthesis by catalysing the conversion of glyceraldehyde-3-phosphate to 1,3-bisphosphoglycerate [70,72]. |
| | | | IPR019734, GO:0005515 | Tetratricopeptide repeat | The TPR motif interacts directly with isoamylase 1 (ISA1), a key enzyme involved in starch synthesis [63] |
| | | | IPR002554, GO:0000159 | Protein phosphatase 2A regulatory B subunit (B56 family) | Interacts directly with isoamylase 1 (ISA1), a key enzyme involved in starch synthesis [73]. |
| | | | IPR006813, GO:0003830 | Glycosyltransferase family 17 | Involved in the modification of carbohydrates by transferring sugar moieties onto specific molecules [65,74]. |
| Chr20—16607018 | DMC | 16107018-17107018 | IPR013766, GO:0015035 | Thioredoxin domain | Thioredoxin serves as a general protein disulphide oxidoreductase [75]. |
| | | | IPR002213, GO:0016758 | UDP-glucuronosyl/UDP-glucosyltransferase | Microsomal enzymes which catalyze the transfer of glucuronic acid to a wide variety of exogenous and endogenous lipophilic substrates [76]. |
| | | | IPR031330, GO:0005975 | Glycoside hydrolase 35, catalytic domain | Enzymes that hydrolyze the glycosidic bond between two or more carbohydrates, or between a carbohydrate and a non-carbohydrate moiety [77]. |
| Chr17_522272 | BldT | 22272-1022272 | IPR001245, GO:0004672 | Protein tyrosine kinase | Protein kinases catalyze the transfer of γ-phosphate from nucleotide triphosphates (often ATP) to one or more amino acid residues in a protein substrate side chain, resulting in a conformational change affecting protein function [51,78]. |
| | | | IPR002213, GO:0016758 | UDP-glucoronosyl and UDP-glucosyl transferase | Microsomal enzymes which catalyze the transfer of glucuronic acid to a wide variety of exogenous and endogenous lipophilic substrates [76]. |
| | | | IPR001128, GO:0005506 | Cytochrome P450 | Oxidoreductase activity, acting on paired donors, with incorporation or reduction of molecular oxygen [47]. |
| Chr19_3840797 | BldT | 3340797-4340797 | IPR019734, GO:0005515 | Tetratricopeptide repeat | The TPR motif consists of 3–16 tandem repeats of 34 amino acid residues, although individual TPR motifs can be dispersed in the protein sequence [79]. |
| Chr14_21319040 | PndT | 20819040-21819040 | IPR001394, GO:0016579 | Ubiquitin carboxyl-terminal hydrolase | Thiol proteases that recognize and hydrolyze the peptide bond at the C-terminal glycine of ubiquitin [80]. |
| | | | IPR000719, GO:0004672 | Protein kinase domain | Protein kinases catalyze the transfer of γ-phosphate from nucleotide triphosphates (often ATP) to one or more amino acid residues in a protein substrate side chain, resulting in a conformational change that affects protein function. Phosphoprotein phosphatases catalyze the reverse process [51]. |
| | | | IPR001245, GO:0004672 | Protein tyrosine kinase | Protein kinases catalyze the transfer of γ-phosphate from nucleotide triphosphates (often ATP) to one or more amino acid residues in a protein substrate side chain, resulting in a conformational change affecting protein function [50,51] |
| | | | IPR008183, GO:0005975 | Aldose 1-epimerase | Enzyme involved in converting glucose-6-phosphate to glucose-1-phosphate. It is used as a substrate for the synthesis of ADP-glucose, the precursor for starch biosynthesis [81]. |
| | | | IPR006652, GO:0005515 | Kelch repeat type 1 | Catalytic unit in galactose oxidase [82]. |

Sixteen putative genes were identified for dry matter content known to play a crucial role in carbohydrate metabolism, starch accumulation, and sucrose synthesis. For example, glycosyl transferase family 29 catalyzes the transfer of sugar moieties to specific acceptor molecules [64]. Li *et al.* [65] confirmed that it is involved in the biosynthesis of complex carbohydrates, including starch and sucrose, by transferring sugar residues to form glycosidic bonds in maize. The remaining candidate genes, such as Serine/threonine-protein kinase, Glycoside hydrolase family 9, and Tetratricopeptide repeat, are involved in carbohydrate metabolism, starch accumulation, and sucrose synthesis in water yam, as also described by Gatarira *et al.* [18].

For boiled and pounded tuber quality, majority of identified candidate genes play a major role in amylose biosynthesis. For example, six putative genes, such as UDP-glucoronosyl and UDP-glucosyl transferase, protein tyrosine kinase, Cytochrome P450, and Tetratricopeptide repeat, were linked with boiled tuber quality. UDP-glucoronosyl and UDP-glucosyl transferase are microsomal enzymes that are known to catalyze the transfer of glucuronic acid to a wide variety of exogenous and endogenous lipophilic substrates [76]. The protein tyrosine kinase catalyzes the transfer of phosphate from ATP to one or more amino acid residues [50]. Both genes (UDP-glucoronosyl and protein tyrosine kinase) were reported by Asfaw *et al.* [23] to be linked with boiled tuber quality in white Guinea yam. For pounded tuber quality, seven putative genes known to play a significant role in amino acid and starch biosynthesis were identified. Asfaw *et al.* [23] reported majority of these genes in white Guinea yam, except aldose 1-epimerase, an enzyme used as a substrate for the ADP-glucose synthesis, the precursor for starch biosynthesis [81]. The number of genes controlling the expression of targeted traits in yams by this study and previous studies confirms that the traits are under polygenic inheritance.

## Conclusion

In this study, 404 water yam genotypes representing four bi-parental mapping populations represented the panel for GWAS analyses using four quality traits. A total of 20 SNPs linked with tuber oxidation, dry matter content, and boiled and pounded yam quality, explaining low to high phenotypic variance, was identified. This is a useful reference catalogue of SNP markers that can be used in water yam breeding as well as in future studies uncovering biological pathways using gene editing techniques. Furthermore, gene annotations identified 35 putative/candidate genes associated with these traits. These results represent an insight into water yam improvement for tuber quality, and further genetic studies are needed to confirm the association as well as the candidate/putative genes displayed in this study.

## Supporting information

**S1 Table. SNP markers associated with tuber oxidative browning in water yam.**
(DOCX)

**S2 Table. SNP markers associated with tuber dry matter content in water yam.**
(DOCX)

**S3 Table. SNP markers associated with tuber boiled quality in water yam.**
(DOCX)

**S4 Table. SNP markers associated with tuber pounded quality in water yam.**
(DOCX)

**S5 Fig. Heatmap LD haplotype blocks of SNP markers located on different chromosomes associated with TOxB.**
The R2 color key indicates the degree of significant association with the putative genes.
(DOCX)

**S6 Fig. Heatmap LD haplotype blocks for different SNP markers located on different chromosomes associated with DMC.** The R2 color key indicates the degree of significant association with the putative genes.
(DOCX)

**S7 Fig. Heatmap LD haplotype blocks for different SNP markers located on different chromosomes associated with BldT.** The R2 colour key indicates the degree of significant association with the putative genes.
(DOCX)

**S8 Fig. Heatmap LD haplotype blocks for different SNP markers located on different chromosomes associated with PndT.** The R2 color key indicates the degree of significant association with the putative genes.
(DOCX)

## Acknowledgments

A special acknowledgment is given to Dr. Jessen V. Bredeson for making the genotyping data available for this study.

## Author contributions

**Conceptualization:** Ranjana Bhattacharjee, Fatoumata Ouattara, Bunmi Olasanmi, Malachy O. Akoroda.

**Data curation:** Fatoumata Ouattara.

**Formal analysis:** Fatoumata Ouattara, Paterne A. Agre, Idris I. Adejumobi.

**Investigation:** Fatoumata Ouattara.

**Methodology:** Ranjana Bhattacharjee, Fatoumata Ouattara, Paterne A. Agre, Idris I. Adejumobi.

**Software:** Fatoumata Ouattara, Paterne A. Agre.

**Supervision:** Ranjana Bhattacharjee, Bunmi Olasanmi, Malachy O. Akoroda.

**Validation:** Ranjana Bhattacharjee, Fatoumata Ouattara, Paterne A. Agre, Idris I. Adejumobi.

**Visualization:** Fatoumata Ouattara, Paterne A. Agre, Idris I. Adejumobi.

**Writing – original draft:** Fatoumata Ouattara.

**Writing – review & editing:** Ranjana Bhattacharjee, Fatoumata Ouattara, Paterne A. Agre, Idris I. Adejumobi, Adekemi Stanley, Fatogoma Sorho, Bunmi Olasanmi, Konan E. B. Dibi, Malachy O. Akoroda.

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
