## [Decision Letter · Decision Letter 0]

20 Oct 2025

Genome-wide association study revealed genomic regions associated with tuber quality traits in water yam (Dioscorea alata L.)

PLOS ONE

Dear Dr. Bhattacharjee,

Thank you for submitting your manuscript to PLOS ONE. After careful consideration, we feel that it has merit but does not fully meet PLOS ONE’s publication criteria as it currently stands. Therefore, we invite you to submit a revised version of the manuscript that addresses the points raised during the review process.

Guidelines for resubmitting your figure iles are available below the reviewer comments at the end of this letter.

We look forward to receiving your revised manuscript.

Kind regards,

Angela T. Alleyne, Ph.D

Academic Editor

PLOS ONE

Journal Requirements:

“The first author is a recipient of PhD research funding from PAN African University of Life & Earth Science Institute (PAULESI) from 2021 to 2024.

Financial support was also obtained from Bill & Melinda Gates Foundation by co-author Paterne A. Agre (BMGF, Grant Number OPP1052998).”

4. In the online submission form you indicate that your data is not available for proprietary reasons and have provided a contact point for accessing this data. Please note that your current contact point is a co-author on this manuscript. According to our Data Policy, the contact point must not be an author on the manuscript and must be an institutional contact, ideally not an individual. Please revise your data statement to a non-author institutional point of contact, such as a data access or ethics committee, and send this to us via return email. Please also include contact information for the third party organization, and please include the full citation of where the data can be found.

Additional Editor Comments:

Provide a more detailed description of the composition of the experimental population and discuss the possible consequences of this population structure for the interpretation of the results

Incorporate a summary sentence or two that includes more detail of significant findings related to genetic traits related to cooking quality for example browning and their chromosome locations in the abstract.  For example, are there any SNP hotspots in the genome? The abstract need to be a bit stringer in summarising the results.

Reviewers' comments:

Reviewer's Responses to Questions

**Comments to the Author**

1. Is the manuscript technically sound, and do the data support the conclusions?

Reviewer #1: Yes

Reviewer #2: Yes

2. Has the statistical analysis been performed appropriately and rigorously?

Reviewer #1: I Don't Know

Reviewer #2: Yes

3. Have the authors made all data underlying the findings in their manuscript fully available?

Reviewer #1: Yes

Reviewer #2: Yes

4. Is the manuscript presented in an intelligible fashion and written in standard English?

Reviewer #1: Yes

Reviewer #2: Yes

Reviewer #1: General comments

The manuscript by Ouattara and co-authors reports a genome wide association study caried on an experimental population of yam (Dioscorea alata L.). The methods seem correct, and the results have the potential to bring valuable insights into the breeding efforts for this specie. The manuscript also appears clearly written, however, some important details are still missing in the method section and should be clarified (see specific comments). I would encourage the authors to provide a more detailed description of the composition of the experimental population and discuss the possible consequences of this population structure for the interpretation of the results.

Specific comments

Line 22: compared

Line 23: This transition from general information to association mapping is a little sudden. Given that the abstract is well under the 300 words limit you could expend it. I would suggest that before “A panel of 404 […]” you could add a phrase explaining the expected benefits from performing association mapping on that specie. Given that the former phrase lists several advantages of Dioscorea alata over other species, one may want to know what trait needs to be improved, and why association mapping is usefull in that context.

Line 38: replace “sensorial” by organoleptic attributes

Line 43, we have already established that yam is a root and tuber crop, the phrase could thus be shortened as “Yams are playing a significant role […]”

Line 50: Remove “natural” : exposure to the [natural] air

Line 51: I suggest the following revision: “associated with modified taste (bitterness) and modified texture (hardness)”

Also please check you word choices, I think Hardness is the correct term for describing how firm a vegetable’s texture is. “Hardiness” refers to a plant’s ability to survive adverse conditions, not to the feel of the fruit itself.

Line 53-55, I suggest the following revision: These preferences have led the end users to favor some yam varieties/species against others, regardless their better agronomic characteristics [REF] such as (pest resistances? Draught resistance? better yield? [REF])

Line 63: anti-inflammatory, anti-cancer, anti-leprosy : each of these claims need a reference

Line 99-100: how were the parents selected? (randomly / On the base of diversity of phenotypes?). What was the number of progeny in each family? In the present version, the information about uneven progeny number between families is only given in the discussion. This should be clearly stated in the method section.

Line 104: “three mature and clean tubers”: does this mean one per replicate?

Line 133-139 (Genotyping methods): Please ellaborate on the sequencing mehtods, what was the length of the sequences? What mapping approach (software, paramters)? Was any other filtering applied to the SNPs besides missing values and MAF (mapping quality? Coverage?)

Line 149-156 (Population structure and GWAS): More details are needed here regarding the determination of the number of clusters. Which method / statistics was applied? Given that 4 populations were used and only three genetic clusters were retained, the methods behind this result need to be clearly described.

Line 165: estimates of adjusted means : adjusted how?

Line 188-189 + table2: “significant difference effects […]” Difference between what?

I don’t understand this part of the results. It looks like a summary statistics of the different phenotypes but I don’t know what the p.values and F values in table 2 come from. What is being tested here? The Methods section mentions variance analysis using lme4 (Line 142-145) but you don’t specify what is the response variable nor what are the predictors.

Line 203-213 (population structure) these results could be more concise. My suggestion would be a phrase reporting the result from estimating the number of clusters, for example: The number of genetic clusters (K) in the population was tested from K=1 to K=X, with K=Y appearing as the most likely with a (insert you statistic here :likelihood / BIC / … ) difference with the second most likely configuration (K=Z). Then summarize the genetic assignment with a table reporting the number of individuals from each population in each cluster

Group1 Group2 Group3 Undetermined (<60% assignment) Admix (group1/goup2) Admix (group1/goup3)

Pop1

Pop2

Pop3

Pop4

If possible, adding the number of admixed individual in each population, and the nature of the admixture (group1/group2; group1/group3…etc) may give you an interesting insight regarding the genetic background of the parents in each population.

L224-258 (GWAS): I am not familiar with the R package you used for the GWAS, but as I understand, it estimates multi-locus random SNP effect with multiple methods (mrMLM, FASTmrEMMA, ISIS EMBLASSO…etc). In table S1 results from all methods are reported but in figure 5 to 8 and in the text of the result section, multiple methods are no longer mentioned. I thus wonder if the results presented in the text are somehow a consensus of the different methods, in such case the approach to compute the consensus should be described. Alternatively, is a SNP considered as significantly associated with a trait as soon as it is significant by one on the methods? In any case, the way the results from multiple methods are compiled or delt with need to be described.

It also appears that GWAS is performed with three different models: one accounting for no population structure (naive), one accounting for population genetic structure (Q) and one accounting for population structure and kinship (Q+K). However, given that the experimental population consists of four family crosses, it is difficult to understand why models without kinship effect are considered. A likelihood comparison between the three models would be welcome.

Line 347-350: this should be stated in the methods

Figure2: I would have liked to see figure2 organized by population on the x axis, clearly separated, so one could see the distribution of individual assignment within each population.

Figure 5-8 : It is not clear why the SNPs over the significance threshold are colored differently (some blue some pink). The figure legend does not explain that use of the colors.

The significance threshold is represented by a dotted line (not solide, as stated in the figure caption).

Reviewer #2: This is a very interesting research topic that can contribute tremendously to food security considering its wide consumption in West Africa and the Caribbean.

Line 74, 91- When using abbreviation, spell out the first phrase GWAS, QTL

Line 306- Pounded

Minor grammar and punctuation revision

**Do you want your identity to be public for this peer review?** For information about this choice, including consent withdrawal, please see our Privacy Policy

Reviewer #1: No

Reviewer #2: **Yes:** Afiya John

---

## [Author Response · Author response to Decision Letter 1]

3 Dec 2025

Point-by-point Response to reviewers’ comments

Reviewer 1

Comment 1: Line 22: compared

Response: We thank the reviewer; the word has been corrected in the manuscript.

Comment 2: Line 23: This transition from general information to association mapping is a little sudden. Given that the abstract is well under the 300 words limit you could expend it. I would suggest that before “A panel of 404 […]” you could add a phrase explaining the expected benefits from performing association mapping on that specie. Given that the former phrase lists several advantages of Dioscorea alata over other species, one may want to know what trait needs to be improved, and why association mapping is useful in that context.

Response: We appreciate the reviewer’s comment. We have included additional information in the abstract to enhance readability and flow as suggested. We added the below text from Line 23-30:

“Despite these promising characteristics, water yam remains less preferred by consumers for traditional food products, particularly boiled and pounded yam preparations. Fast and efficient development of superior genotypes that meet farmers and end-users needs have been challenging through classical breeding methods. The objective of the study was to use genome-wide associations to assess the genetics of post-harvest tuber quality, mainly targeting the consumer-preferred traits. A panel of 404 water yam genotypes were assessed to decipher the genomic regions associated with traits such as”

Comment 3: Line 38: replace “sensorial” by organoleptic attributes

Response: We thank the reviewer for this comment “sensorial” has been replaced by “organoleptic” attributes as suggested (Line 47).

Comment 4: Line 43, we have already established that yam is a root and tuber crop, the phrase could thus be shortened as “Yams are playing a significant role […]”

Response: We appreciate this comment from the reviewer. We shortened the phrase as suggested (Line 53)

Comment 5: Line 50: Remove “natural” : exposure to the [natural] air

Response: “natural” has been removed as suggested (Line 60).

Comment 6: Line 51: I suggest the following revision: “associated with modified taste (bitterness) and modified texture (hardness)”

Also please check you word choices, I think Hardness is the correct term for describing how firm a vegetable’s texture is. “Hardiness” refers to a plant’s ability to survive adverse conditions, not to the feel of the fruit itself.

Response: We thank the reviewer for the suggestion. The sentence has been corrected as suggested (Line 61 – Line 62).

Comment 7: Line 53-55, I suggest the following revision: These preferences have led the end users to favor some yam varieties/species against others, regardless their better agronomic characteristics [REF] such as (pest resistances? Draught resistance? better yield? [REF])

Response: We appreciate the reviewer and made necessary changes as suggested (Line 63 – Line 66).

Comment 8: Line 63: anti-inflammatory, anti-cancer, anti-leprosy : each of these claims need a reference

Response: We have added suitable references as suggested by the reviewer.

Comment 9: Line 99-100: how were the parents selected? (randomly / On the base of diversity of phenotypes?). What was the number of progeny in each family? In the present version, the information about uneven progeny number between families is only given in the discussion. This should be clearly stated in the method section.

Response: We appreciate the suggestion of the reviewer and have provided all relevant information regarding the parents and number of individuals per family in the revised version of the manuscript. The added text is as below from Line 110-Line 128:

“A panel of 404 D. alata genotypes with diverse agronomic and tuber quality attributes were selected for the present study (Table 1). These genotypes were selected from the pool of progenies of four different bi-parental populations (using four female and one male parents) developed by Yam Improvement Program (YIP) of the International Institute of Tropical Agriculture (IITA), Ibadan, Nigeria. The four bi-parental populations were combined to carry out GWAS analyses since the population size of each mapping population was not big enough (Table 1) to make-up the diversity panel. The parents were selected based on their breeding values for traits including tuber yield, dry matter content, and resistance to anthracnose disease. The GWAS panel therefore consisted of 196 progenies and two parents making up the first family/population, 103 progenies and two parents for the second family/population, 54 progenies and two parents for the third family/population, and 43 progenies and two parents for the fourth family/population (Table 1). For field establishment, three mini-setts (approximately 50 g) of each genotype were planted on a 3m one-row plot in an alpha lattice design representing three replications at IITA, Ibadan (7°40’19.62” N, 3°91’73.13” E, and 189 m above sea level), Nigeria.”

We also added the following information as footnote of Table 1 for better clarity on the number of genotypes used for analyses (Line 156 – Line 162):

“Note: The male parent was common across the four populations. The first two digits in the parental IDs denote the year of crossing. All crosses were performed at IITA. Although the total number of parents used for generating the crosses were five (four female and one male), however, for analyses we considered the sequence information of male parent independently for each population (SNP data was different across each male parent for each population, which could be attributed to the clonal maintenance of the crop), resulting in eight parents (four female and four male) and a total panel of 404 genotypes.”

Comment 10: Line 104: “three mature and clean tubers”: does this mean one per replicate?

Response: Yes, for each genotype, one mature, clean tuber was selected from each replicate. We added the information in the text (Line 130).

Comment 11: Line 133-139 (Genotyping methods): Please ellaborate on the sequencing mehtods, what was the length of the sequences? What mapping approach (software, paramters)? Was any other filtering applied to the SNPs besides missing values and MAF (mapping quality? Coverage?)

Response: We thank the reviewer for the valuable suggestions. We have provided additional information while the information on length of the sequences and the mapping approach used are described in Bredeson et al. (2022). We have made necessary changes as follows and presented in Line 167 – Line 182:

“DNA from each genotype was isolated at IITA using modified CTAB methods and were genotyped at the Integrated Genotyping Service and Support (IGSS, BecA-ILRI hub, Nairobi, Kenya) using the ‘high-density’ DArTseq reduced-representation method. The details of sequence length, mapping approach (software used) and other parameters are provided in Bredesson et al [19].

DArTseq genotyping datasets were thereafter mapped onto the v2 genome sequence [17] of water yam (479.5 Mb, 20 chromosomes, ~25,189 protein-coding genes). The raw genotyping data consisted of 19,012 SNPs for family 1 (TDa1419), 17,126 SNPs for family 2 (TDa1427), 13,857 SNPs for family 3 (TDa1403), and 26,643 SNPs for family 4 (TDa1402). The common set of SNPs across the four populations were then searched and a total of 4,805 SNPs were identified for further analyses. The 4,805 SNPs were subjected to quality control using the parameters of missing values > 20% and minor allele frequency (MAFs) < 0.05, heterozygosity and unmapped SNPs, which resulted in 4,646 high-quality SNPs distributed across the 20 chromosomes of water yam.”

Comment 12: Line 149-156 (Population structure and GWAS): More details are needed here regarding the determination of the number of clusters. Which method / statistics was applied? Given that 4 populations were used and only three genetic clusters were retained, the methods behind this result need to be clearly described.

Response: We thank the reviewer for this critical observation. In the revised manuscript, we have expanded the Methods section on population structure analysis to clearly describe how the number of clusters was determined and other details. The added information is from Line 196 – Line 218 in the revised manuscript:

“To assess the genetic stratification within water yam panel and the relationship among genotypes, population structure analysis was conducted using three complementary approaches including ADMIXTURE, principal component analysis (PCA), and phylogenetic clustering. To determine the optimal number of genetic cluster (K), ADMIXTURE was run for K value ranging from 1 to 10, and the most appropriate K was identified using the cross-validation (CV) error method implemented in ADMIXTURE model from the LEA package in R [36]. The K value with the lowest CV error was considered the best-supported model. Genotypes were assigned to clusters when their ADMIXTURE membership probability (MP) was ≥ 0.60, and those below this threshold were considered admixed. PCA was performed using factoMiner [37] and factoExtra [38] packages to visualize the distribution of genotypes along major axes of genetic variation, and the clustering pattern from PCA was compared with ADMIXTURE results for consistency. In addition, hierarchical clustering based on pairwise genetic distances (APE package) implemented in R [39; 40] was used to further validate the number and membership of clusters.

From the ADMIXTURE model, the Q matrix was obtained. Further, rrBLUP package [39] in R was used to generate the relationship matrix as an important covariate in the GWAS model. Finally, the GWAS analysis was conducted using multi-locus random-SNP-effect mixed linear model (mrMLM) in mrMLM package [40] implemented in R. The GWAS model was based on the equation 4:”

Comment 13: Line 165: estimates of adjusted means : adjusted how?

Response: We thank the reviewer for this observation. By “adjusted means” we meant the combined BLUEs from the two years of evaluation. We have now changed the “adjusted means” to BLUEs in the revised manuscript (Line 228).

We also provided additional information from Line 236 – Line 241 towards model fit and to address the Comment 16 of the reviewer:

“The model fit was further compared using Akaike Information Criterion/Bayesian Information Criterion (AIC/BIC) to identify which model consistently provided the best fit across traits. To ensure robust detection of trait-marker associations, a SNP was considered significant only when it exceeded the threshold LOD (FDR-adjusted), and equally identified by at least two of the six multi-locus methods under the same correction model.”

Comment 14: Line 188-189 + table2: “significant difference effects […]” Difference between what?

I don’t understand this part of the results. It looks like a summary statistics of the different phenotypes but I don’t know what the p.values and F values in table 2 come from. What is being tested here? The Methods section mentions variance analysis using lme4 (Line 142-145) but you don’t specify what is the response variable nor what are the predictors.

Response:

We appreciate the reviewer’s observation regarding clarity. In the revised manuscript, we have replaced the summary statistics table with a proper ANOVA mean squares table (new Table 3) and enhanced the interpretation of the results to improve overall clarity (Line 256 – Line 257). We also made changes in the Materials and Methods section accordingly (Line 186 – Line 189).

Comment 15: Line 203-213 (population structure) these results could be more concise. My suggestion would be a phrase reporting the result from estimating the number of clusters, for example: The number of genetic clusters (K) in the population was tested from K=1 to K=X, with K=Y appearing as the most likely with a (insert you statistic here :likelihood / BIC / … ) difference with the second most likely configuration (K=Z). Then summarize the genetic assignment with a table reporting the number of individuals from each population in each cluster

Group1 Group2 Group3 Undetermined (<60% assignment) Admix (group1/goup2) Admix (group1/goup3)

Pop1

Pop2

Pop3

Pop4

If possible, adding the number of admixed individual in each population, and the nature of the admixture (group1/group2; group1/group3…etc) may give you an interesting insight regarding the genetic background of the parents in each population.

Response:

We appreciate the reviewer immensely for the guidance. This has surely allowed us to present the results in a concise manner. We have added a Table (Table 4) and adjusted the text description accordingly and as below (Line 275 – Line 284):

“Based on CV error curve, clearly indicated that K = 3 had represented the minimum prediction error, and therefore The population structure analysis revealed three genetic groups clusters were retained (Table 4; Fig 2). These genetic clusters comprised 296 water yam genotypes representing (73.27% of the total number of genotypes under study and remaining 108 genotypes (26.73%) were admixed.) The details of cluster composition and admixed genotypes is presented in Table 4 and Figure 2. Cluster 1 didn’t represent any genotypes or admixed population from Family 1 and Family 2, although 5.7% of genotypes from Family 2 were admixed. The remaining two clusters represented genotypes and admixed populations from all four families (Table 4).”

Comment 16: L224-258 (GWAS): I am not familiar with the R package you used for the GWAS, but as I understand, it estimates multi-locus random SNP effect with multiple methods (mrMLM, FASTmrEMMA, ISIS EMBLASSO…etc). In table S1 results from all methods are reported but in figure 5 to 8 and in the text of the result section, multiple methods are no longer mentioned. I thus wonder if the results presented in the text are somehow a consensus of the different methods, in such case the approach to compute the consensus should be described. Alternatively, is a SNP considered as significantly associated with a trait as soon as it is significant by one on the methods? In any case, the way the results from multiple methods are compiled or delt with need to be described.

It also appears that GWAS is performed with three different models: one accounting for no population structure (naive), one accounting for population genetic structure (Q) and one accounting for population structure and kinship (Q+K). However, given that the experimental population consists of four family crosses, it is difficult to understand why models without kinship effect are considered. A likelihood comparison between the three models would be welcome.

Response: We appreciate the reviewer's observation and comments. We thank the reviewer for the insightful comments regarding the integration of results from multiple GWAS methods and the choice of naïve, Q, and Q+K models.

i. Use of multiple GWAS methods and how results were combined:

In the revised manuscript, we have explicitly stated that a SNP was declared significantly associated with a trait only when it was detected by at least two of the six multi-locus methods under the same correction model (naive, Q, or Q+K) (Line 233 – Line 238). This provides a conservative consensus strategy and minimizes false positives inherent to single-method detection. We have clarified this procedure in both the Methods and Results sections (Line 312 – Line 313). We used the consolidated results for demonstration in the figures rather than the outputs of individual methods.

ii. Use of naive, Q, and Q+K models:

We agree with the reviewer that kinship is expected to be important in a population consisting of four related family groups. To address this, we now provide a likelihood comparison (AIC/BIC) among the naïve, Q, and Q+K models to demonstrate that the Q+K model provided the best fit for all traits, thereby supporting its primary use in interpreting significant associations. The naïve and Q-only models were retained to illustrate the inflation of false positives when kinship is ignored and to maintain comparability with previous multi-locus GWAS studies.

We have added the information in GWAS subsection of Results (Line 318 – Line 319) and Discussion sec

---

## [Decision Letter · Decision Letter 1]

15 Dec 2025

Genome-wide association study revealed genomic regions associated with tuber quality traits in water yam (Dioscorea alata L.)

PONE-D-25-46993R1

Dear Dr. Bhattacharjee,

We’re pleased to inform you that your manuscript has been judged scientifically suitable for publication and will be formally accepted for publication once it meets all outstanding technical requirements.

Kind regards,

Angela T. Alleyne, Ph.D

Academic Editor

PLOS One

---

## [Editor Report · Acceptance letter]

PONE-D-25-46993R1

PLOS One

Dear Dr. Bhattacharjee,

I'm pleased to inform you that your manuscript has been deemed suitable for publication in PLOS One. Congratulations! Your manuscript is now being handed over to our production team.

Kind regards,

on behalf of

Dr. Angela T. Alleyne

Academic Editor

PLOS One